# Single-Step Consistent Diffusion Samplers

## Abstract

Sampling from unnormalized target distributions is a fundamental yet challenging task in machine learning and statistics. Existing sampling algorithms typically require many iterative steps to produce high-quality samples, leading to high computational costs that limit their practicality in time-sensitive or resource-constrained settings. In this work, we introduce *consistent diffusion samplers*, a new class of samplers designed to generate high-fidelity samples in a single step. We first propose Consistency-Distilled Diffusion Samplers (CDDS), which demonstrates that consistency distillation can be accomplished within sampling contexts in the absence of pre-collected training datasets. To eliminate the need for a pre-trained sampler, we further propose Self-Consistent Diffusion Samplers (SCDS), which performs self-distillation during training. SCDS learns to perform diffusion sampling and to skip intermediate steps via a self-consistency loss. Through extensive experiments on a variety of synthetic and real-world unnormalized distributions, we show that our approaches yield high-fidelity samples using less than 1% of the network evaluations required by traditional diffusion samplers.

## 1 Introduction

Sampling from densities of the form

$$p_{\text{target}} = \frac{\rho}{Z}, \quad \text{with} \quad Z = \int_{\mathbb{R}^D} \rho(X) \mathrm{d}X \tag{1}$$

with $\rho$ evaluable pointwise but $Z$ intractable, is a central problem in machine learning [21, 30] and statistics [3, 31], and has applications in scientific fields like physics [1, 33, 53], chemistry [17, 23, 24], and many other fields involving probabilistic models.

Many established sampling algorithms are inherently iterative, with the accuracy of the final samples depending heavily on the number of steps. Classical Markov chain Monte Carlo (MCMC) methods asymptotically converge to the target distribution with an infinite number of steps [27, 39]. Nonetheless, the finite number of feasible steps in real-world applications means that MCMC can only provide an approximation and cannot guarantee an exact solution. As an improvement, more recent diffusion-samplers [7, 50, 55] guarantee convergence in a finite number of steps. However, they often necessitate hundreds of iterations to yield high-quality samples. Such iterative samplers tend to suffer from slow mixing, making them still impractical for use in large models and resource-limited scenarios.

From another perspective, recent work on diffusion generative models [22, 44, 47, 48] has demonstrated the feasibility of fewer-step sampling. This can be achieved using knowledge distillation [41, 46] or consistency training [46], potentially enabling even single-step generation. However, directly applying the distillation techniques to distillate existing diffusion samplers is challenging, as it often requires large datasets of samples that are expensive to collect in practice. Moreover, the

learning targets of diffusion samplers and consistency training are conflicted in some aspects. These make reducing the sampling steps of diffusion samplers a challenging problem.

In this paper, we propose *consistent diffusion samplers* to produce high-quality samples in a single step. We first show that diffusion-based samplers can be *consistently distilled* into single-step diffusion samplers and propose the Consistency-Distilled Diffusion Samplers (CDDS) approach. Instead of storing a large dataset of fully diffused samples, CDDS exploits incomplete trajectories and noisy samples encountered during the diffusion process, hence reducing the unnecessary costs. We further introduce the Self-Consistent Diffusion Sampler (SCDS) method that does not require a pretrained diffusion sampler. Instead, it fully amortizes exploration by jointly learning both diffusion sampling and large cut off steps that match the outcome of paths of small steps. This enables single-step sampling yet retains the option to refine samples through multiple iterations if desired, subsuming existing diffusion-based approaches. Our contributions can be summarized as follows:

- We show that diffusion-based samplers for unnormalized distributions can be effectively distilled into single-step consistent samplers without pre-collecting large datasets of samples.

- We introduce a self-consistent diffusion sampler that learns to perform single-step sampling by jointly training diffusion-based transitions and large shortcut steps via a self-consistency criterion. This method only trains one neural network and does not require pretrained samplers or high-quality datasets.

- Through extensive evaluations on synthetic and real unnormalized distributions, we demonstrate that our method delivers competitive sample quality while drastically reducing sampling steps.

## 2  Related Work

**Monte Carlo-based Samplers.**   Monte Carlo-based Samplers, such as Markov chain Monte Carlo (MCMC), are a classical approach for sampling from unnormalized target densities. The key idea is to construct a Markov chain whose stationary distribution matches the target distribution [10]. Prominent examples include the Metropolis-Hastings algorithm [20, 29], Gibbs sampling [19], and Langevin dynamics [35, 40]. By exploiting geometric structure in the target distribution, Hamiltonian Monte Carlo [10, 12, 14, 27] often leads to more efficient exploration. To address scalability challenges in high-dimensional or large-dataset scenarios, stochastic gradient MCMC variants [12, 52, 56, 57] have been introduced. Although these MCMC methods reduce per-step computational costs or improve mixing, they remain inherently iterative, requiring many transitions to yield high-quality samples.

**Diffusion-based Samplers.**   An alternative viewpoint frames sampling as an optimal control task [7, 37, 38, 55], where controlled stochastic differential equations transport an initial distribution to the target via a Schrödinger bridge [42, 43]. This approach has recently motivated numerous diffusion-based sampling methods [11, 13, 18, 36, 50]. Further improvements have been explored through Hamiltonian dynamics [8], intermediate resampling strategies [11], and physics-informed neural networks to evolve densities [49]. Recent methods such as CMCD [51] jointly optimize forward and backward diffusion dynamics. Additionally, theoretical connections between GFlowNets [4, 5] and diffusion-based sampling have been investigated [6, 54]. For an extensive review of relevant metrics and baseline samplers, see [9]. Current diffusion methods partially amortize sampling costs in training but still require iterative inference-time generation. In this work, we fully amortize exploration during training, enabling efficient single-step sampling at inference.

**Consistent Generative Models.**   Recent work in generative modeling has introduced the notion of consistency. Consistency models [26, 45, 46] learn direct mapping from any intermediate state to the terminal state. Progressive distillation [28, 41] incrementally distills a trained diffusion model into a more efficient version that takes half as many steps. Shortcut models [16] apply this distillation principle during training, enabling direct learning of efficient transitions without relying on a pretrained teacher. We extend this line of work to the setting of sampling from unnormalized densities, assuming only pointwise access to the target density $\rho$, without requiring data samples.

## 3 Preliminaries: Sampling via Controlled Stochastic Processes

Diffusion samplers aim to draw samples from a complex target density $p_{\text{target}} = \rho/Z$ by transporting them from a simpler prior density $p_{\text{prior}}$. We consider forward and reverse-time stochastic processes on $\mathbb{R}^d$ over a time interval $[0, T]$, each described by the following SDEs:

$$\mathrm{d}X_s = (\mu + \sigma u)(X_s, s)\mathrm{d}s + \sigma(s)\mathrm{d}W_s, \quad X_0 \sim \pi, \tag{2}$$

$$\mathrm{d}X_s = (-\mu + \sigma v)(X_s, T-s)\mathrm{d}s + \sigma(T-s)\mathrm{d}W_s, \quad X_0 \sim \tau, \tag{3}$$

where $u, v \in \mathcal{U} \subset C(\mathbb{R}^d \times [0, T], \mathbb{R}^d)$ are control functions, $\mu \in C(\mathbb{R}^d \times [0, T], \mathbb{R}^d)$ is a linear drift, $\sigma \in C([0, T], \mathbb{R})$ is the diffusion coefficient, and $\mathrm{d}W_s$ denote forward Brownian increments. We seek $u$ and $v$ such that (2) and (3) become time-reversal counterparts.

Let $\mathbb{P}^{u,\pi}$ and $\overline{\mathbb{P}}^{v,\tau}$ be the path measures induced by (2) and (3), respectively. Consider a divergence $D : \mathcal{P} \times \mathcal{P} \to \mathbb{R}_{\geq 0}$. We aim to solve the optimization problem:

$$u^*, v^* \in \arg\min_{u,v\in\mathcal{U}} D(\mathbb{P}^{u,\pi} \mid \overline{\mathbb{P}}^{v,\tau}). \tag{4}$$

When the divergence $D$ reaches its minimum of zero in (4), the marginal distribution at terminal time matches exactly, $\mathbb{P}_T^{u,\pi} = \tau$. Consequently, by selecting $\pi = p_{\text{prior}}$ and $\tau = p_{\text{target}}$, one can generate samples from the target distribution $p_{\text{target}}$ through simulating (2) with the optimal control $u^*$.

Nelson's identity provides a local characterization of optimality conditions [2, 15, 32]. It states that the forward path measure $\mathbb{P}^{u,\pi}$ coincides with the reverse-time measure $\overline{\mathbb{P}}^{v,\tau}$ if and only if the forward drift can be expressed as the backward drift adjusted by the scale score $u(\cdot, s) = v(\cdot, s) + \sigma(s)\nabla \log \mathbb{P}_s^{u,\pi}(\cdot)$. In practice, the marginal distributions $\mathbb{P}^{u,\pi}s$ are generally intractable. Therefore, we typically approximate solutions to (4) by parameterizing the control $u$ with a neural network $u_\theta$. Training these models typically proceeds through the following iterative steps:

1. Simulate a batch of $M$ trajectories $\{(X_s^{(i)})_{0\leq s\leq T}\}$, $i = 1, \ldots, M$, using the generative process (2).

2. Compute the divergence measure and its gradient with respect to the parameters $\theta$.

3. Update the parameters $\theta$ accordingly, and repeat the process until convergence.

The Kullback-Leibler (KL) [7, 50, 55] and the log-variance (LV) divergences are common choices [6, 34, 38]:

$$D_{\text{KL}}(\mathbb{P} \mid \mathbb{Q})(X) = \mathbb{E}\left[\log \frac{\mathrm{d}\mathbb{P}}{\mathrm{d}\mathbb{Q}}(X)\right] + \log Z, \quad D_{\text{LV}}(\mathbb{P} \mid \mathbb{Q})(X) = \mathbb{V}\left[\log \frac{\mathrm{d}\mathbb{P}}{\mathrm{d}\mathbb{Q}}(X)\right]. \tag{5}$$

The likelihood ratio appearing in (5) is given explicitly by the Radon-Nikodym derivative:

$$\log \frac{\mathrm{d}\mathbb{P}^{u,\pi}}{\mathrm{d}\overline{\mathbb{P}}^{v,\tau}} = \int_0^T (u+v)\cdot\left(u_\theta + \frac{v-u}{2} + \nabla\cdot(\sigma v - \mu)\right)\mathrm{d}s + \int_0^T (u+v)\mathrm{d}W_s + \log\frac{p_{\text{prior}}(X_0^\theta)}{p_{\text{target}}(X_T^\theta)} \tag{6}$$

where $X^\theta$ is the trajectory obtained by simulating the forward SDE (2) using the parameterized control $u_\theta$. The log normalization constant from the target density disappears upon taking gradients, making this a practical objective for training. See [7] and Appendix A.2 of [38] for detailed derivations.

Once trained, the optimized control $u_\theta$ allows generation of samples from $p_{\text{target}}$ through forward simulations of (2). In practice, this continuous-time process must be discretized into finite steps $0 = t_1 < t_2 < \cdots < t_N = T$, introducing a trade-off between computational cost and accuracy.

## 4 Consistency Distilled Diffusion Samplers

In this section, we present the *consistency distilled diffusion sampler* (CDDS) method to solve the problem of single-step sampling from unnormalized densities by distillating from a pretrained sampler. Concretely, our goal is to learn a consistency function $f : (X_t, t) \mapsto X_T$, which maps any intermediate state $X_t$ directly to a sample $X_T$ from the target distribution.

A straightforward method is to first appoximate a dataset by simulating the generative SDE in (2) and producing samples $\{\hat{X}_T^i\}_{i=1}^M$, and then applying existing consistency distillation or consistency

| **Algorithm 1** CDDS Training | **Algorithm 2** SCDS Training |
|---|---|
| **Input:** Model parameters $\theta$, a pre-trained control $u$, learning rate $\eta$
$\theta' \leftarrow \theta$
**repeat**
    Sample $X_0 \sim p_{\text{prior}}$ and $n \sim \mathcal{U}\{1, N-1\}$
    Simulate PF ODE of (2) to get $\hat{X}_{t_n}$ and $\hat{X}_{t_{n+1}}$
    $\mathcal{L}(\theta, \theta') \leftarrow \|f_{\theta'}(\hat{X}_{t_{n+1}}, t_{n+1}), f_\theta(\hat{X}_{t_n}, t_n)\|_2$
    $\theta \leftarrow \theta - \eta \, \nabla_\theta \mathcal{L}(\theta, \theta'; u)$
    $\theta' \leftarrow \text{stopgrad}(\theta)$
**until** convergence | **Input:** Model parameters $\theta$, weights $\lambda_{\text{S}}, \lambda_{\text{SC}}$
$\theta' \leftarrow \theta$
**repeat**
    Sample $X_0 \sim p_{\text{prior}}$, $d$, and $t$
    Simulate (2) to get $X_{0:T}$
    Compute target $X_{t+2d}$ from (10)
    Compute shortcut $\hat{X}_{t+2d}$ from (9)
    Compute sampling loss $\mathcal{L}_{\text{S}}$ via (5)
    Compute consistency loss $\mathcal{L}_{\text{SC}}$ via (11)

    $\theta \leftarrow \nabla_\theta \left(\lambda_{\text{S}} \mathcal{L}_{\text{S}} + \lambda_{\text{SC}} \mathcal{L}_{\text{SC}}\right)$
    $\theta' \leftarrow \text{stopgrad} \, \theta$
**until** convergence |

training methods [46] to learn the function $f$. However, this approach is expensive as it necessitates pre-collecting and storing a large dataset. Moreover, the accumulation of numerical errors arises from the numerical solver , resulting in significant global error.

To solve the problem, we propose to leverage intermediate states of the pretrained model during each training iteration. Using these multiple and short intervals among intermediate helps keep the overall global error small.

During model training, we minimize the discrepancy between the outputs of the consistency function at the consecutive intermediate states of the probability flow ODE [48] associated with (2):

$$\mathcal{L}_{\text{CD}}(\theta, \theta'; u)(X) := \mathbb{E}\left[\|f_{\theta'}(\hat{X}_{t_{n+1}}, t_{n+1}), f_\theta(\hat{X}_{t_n}, t_n)\|_2\right], \tag{7}$$

where the expectation is over discrete time indices $n$ and $\theta' = \text{stopgrad}(\theta)$ indicates gradient stopping on the target term. Notably, unlike standard consistency generative models, the states $\hat{X}_{t_{n+1}}$ and $\hat{X}_{t_n}$ are obtained from partial integrations of the probability flow ODE rather than from real data samples. Consequently, training CDDS incurs computational costs similar to training traditional diffusion samplers while substantially accelerating inference. The training procedure is summarized in Algorithm 1.

If the loss (7) is driven to zero, the learned consistency function recovers the true mapping of the probability flow ODE, implying that CDDS can achieve arbitrarily accurate single-step sampling in the limit of sufficiently small integration steps. We formally state this in Theorem 1.

**Theorem 1.** *Let $f_\theta(X_t, t)$ be a consistency function parameterized by $\theta$, and let $f(X_t, t; u)$ denote the consistency function of the PF ODE defined by the control $u$. Assume that $f_\theta$ is $L-$Lipschitz continuous. Additionally, assume that for each step $n \in \{1, 2, \ldots, N-1\}$, the ODE solver called at $t_n$ has a local error bounded by $O((t_{n+1} - t_n)^{p+1})$ for some $p \geq 1$. If $\mathcal{L}_{CD}(\theta, \theta'; u) = 0$, then:*

$$\sup_{n, X_{t_n}} \|f_\theta(X_{t_n}, t_n) - f(X_{t_n}, t_n; u)\|_2 = O((\Delta t)^p), \tag{8}$$

*where $\Delta t := \max_{n \in \{1, 2, \ldots, N-1\}} |t_{n+1} - t_n|$.*

A complete proof is provided in the Appendix. This theoretical result shows that consistency functions can be distilled from diffusion samplers when only an unnormalized density oracle is available, enabling principled single-step sampling without requiring access to data from the target distribution.

**Remark.** While our distillation approach builds upon the core principles of consistency generative models, it differs in setting and requirements. Instead of relying on having access to a dataset from $p_{\text{target}}$, our method extends consistency distillation to sampling from unnormalized distributions, making it applicable beyond generative modeling tasks.

## 5    Self–Consistent Diffusion Samplers

### 5.1    Overiview

Even though CDDS demonstrates that single-step sampling is feasible, it requires a pretrained diffusion sampler for distillation. In this section, we further introduce *self-consistent diffusion sampler* (SCDS) that achieves single-step sampling without requiring the pre-trained diffusion sampler. The key idea is to adapt consistency training into the diffusion-based sampler training. However, the challenge arises from merging two perspectives: diffusion-based samplers learn a time-dependent control function that steers an SDE from a simple prior distribution to the target distribution. Typically, the control is trained on a fixed schedule (e.g., $N$ small increments of length $T/N$ along a discretized time axis), requiring multiple steps. In contrast, consistency models learn a direct mapping from any intermediate state on an ODE to the terminal state. In other words, at time $t$ the model is implicitly taught to jump a large step of length $T - t$. Crucially, consistency training in the original formulation [45, 46] assumes the availability of intermediate states generated by perturbing real data; in the context of sampling from unnormalized densities, however, we cannot generate these states directly because we lack data and thus must learn both sampling and self-distillation simultaneously. Furthermore, as discussed in Section 3, diffusion-based samplers typically parameterize the control function, whose purpose differs fundamentally from that of the consistency function used in standard consistency training.

To reconcile these perspectives and overcome this challenge, we propose conditioning the control function $u_\theta(X_t, t, d)$ on both the current time $t$ and the desired step size $d$. By adjusting $d$, the model can adapt between short incremental steps (as in standard diffusion samplers) and large jumps (as in consistency models). This design amortizes the learning of both small and large transitions into one network and recovers consistency models' single-step sampling by setting $d = T - t$ and diffusion sampling by setting $d = T/N$. In doing so, we avoid training with two contrasting learning targets, hence making it feasible to train a single-step diffusion sampler from scratch.

### 5.2    Training

**Learning the base case $\mathbf{d = T/N}$.**    In standard generative modeling scenarios (where a dataset is available), the base case $d = T/N$ can be learned directly from data using deterministic trajectories [25, 16]. These trajectories provide explicit guidance toward high-density regions of the target distribution.

However, when working with an unnormalized density, a key challenge is discovering high-probability regions [11]. Thus, the process (2) is particularly well-suited for learning the base case as the Brownian motion helps probe different parts of the space, allowing the model to learn and adapt itself to the target distribution. By optimizing $u_\theta(X_t, t, d = T/N)$ under (5), we ensure that the model can generate meaningful transitions from the prior to these regions of interest, forming a strong foundation for self-consistent learning at larger step sizes.

**Enforcing self-consistency.**    To ensure that the step-size-conditioned control function $u_\theta(X_t, t, d)$ remains accurate across varying step sizes, we introduce a self-consistency loss. The key idea is that taking a large step should yield the same result as taking multiple smaller steps. To do so, we impose a consistency condition on the Euler discretization of the probability flow ODE associated with the forward process (5). Specifically, we require that a single large step of size $2d$,

$$\hat{X}_{t+2d} = X_t + 2d \left( \mu + \tfrac{1}{2}\sigma u_\theta \right) (X_t, t, 2d), \tag{9}$$

yields the same result as two smaller steps of size $d$. The intermediate state after the first small step is computed as

$$X_{t+d} = X_t + d \left( \mu + \tfrac{1}{2}\sigma u_{\theta'} \right) (X_t, t, d),$$

and the final state after the second small step is

$$X_{t+2d} = X_{t+d} + d \left( \mu + \tfrac{1}{2}\sigma u_{\theta'} \right) (X_{t+d}, t+d, d), \tag{10}$$

where $\theta' = \mathrm{stopgrad}(\theta)$. The self-consistency objective is a simple least square minimization problem:

$$\mathcal{L}_{\mathrm{SC}} = \mathbb{E} \left[ \| X_{t+2d} - \hat{X}_{t+2d} \|^2 \right], \tag{11}$$

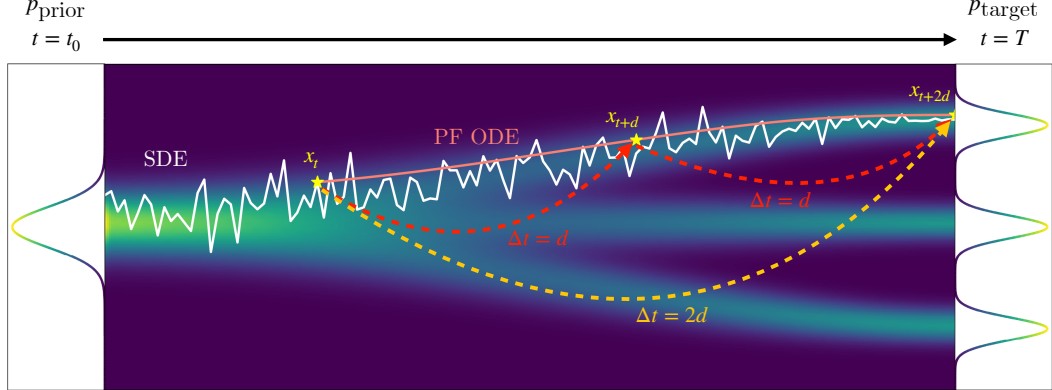

Figure 1: Graphical illustration of the training procedure for SCDS over the path space. First, the SDE (2) is simulated (white) to optimize (5). Next, a timestep $t$ and a step size $d$ are randomly sampled. From $X_t$ on the simulated SDE trajectory, we execute two consecutive steps of size $d$ (red) along the probability flow ODE trajectory (pink) of (2), obtaining the target $X_{t+2d}$. Finally, the large step of size $2d$ (orange) predicts $\hat{X}_{t+2d}$ directly from $X_t$, and the self-consistency loss (11) minimizes the squared difference between $\hat{X}_{t+2d}$ and the two-step target $X_{t+2d}$, ensuring multi-scale consistency.

where the expectation is taken over time indices and step sizes drawn from the simulated trajectories.

This loss encourages the model to correct for numerical errors when taking large steps, allowing it to "skip" multiple smaller steps while remaining consistent with the dynamics of the probability flow ODE.

**End-to-end training algorithm.** As abovementioned, our training procedure jointly optimizes two objectives: (i) a sampling loss (5) for the base case $d = T/N$, which ensures exploration and score approximation by simulating the SDE (2), and (ii) the self-consistency loss (11) enforced on the probability flow ODE of (2) for larger steps, which enforces consistency across multiple time scales. This end-to-end formulation enables the model to fully amortize the cost of sampling into a single forward pass at inference time.

To enable the recursive halving of steps, we discretize the time interval $[0, T]$ into $N + 1$ points, where $N$ is chosen as a power of two. The sampling loss is computed by simulating the forward SDE along this time grid.

For self-consistency training, we sample step sizes $d$ and times $t$ such that $d$ are powers of two (multiplied by $T/N$) dividing the remaining time $T - t$. This ensures that from any time $t$, we can take exactly $k$ steps of size $d$ to reach the terminal state for some integer $k$. This way, training focuses on time sequences that are applicable during inference.

To compute the self-consistency loss, we extract $X_t$ from the simulated forward SDE. Using $X_t$ and the sampled step size $d$, we compute the shortcut step $\hat{X}_{t+2d}$ using (9) and the two-step target trajectory $X_{t+2d}$ using (10). We then optimize their squared difference via the objective (11), ensuring that larger steps remain consistent with fine-grained trajectories. The training procedure is summarized in Algorithm 2 and illustrated in Figure 1. Compared to standard diffusion-based samplers, which typically require hundreds of evaluations per iteration during training, our method introduces only 3 additional function evaluations. Thus, the overhead from SCDS training is marginal, typically amounting to less than a few percent of the total computational cost per iteration.

## 5.3 Inference.

**Few-step sampling.** With a well-trained control $u_\theta$, sampling can be performed in a single step by drawing from the prior and applying a single Euler update with step size $d = T$, as shown in Algorithm 3. This accelerates generation compared to traditional diffusion-based samplers. Alternatively, our method provides a flexible tradeoff between computational efficiency and sample

| **Algorithm 3** Single-Step Sampling with SCDS | **Algorithm 4** Multi-Step Sampling with SCDS |
|---|---|
| **Input:** Trained model $u_\theta$ 
 Sample $X_0 \sim p_{\text{prior}}$ 
 $X_T \leftarrow X_0 + T\left(\mu + \frac{1}{2}\sigma u_\theta\right)(X_0, 0, T)$ 
 **Return** $X_T$ | **Input:** Trained model $u_\theta$, number of sampling steps $N$ 
 Sample $X_0 \sim p_{\text{prior}}$ 
 Initialize $d \leftarrow T/N$ and $t \leftarrow 0$ 
 **for** $k = 1, \ldots, N$ **do** 
 $\quad X_{t+d} \leftarrow X_t + d\left(\mu + \frac{1}{2}\sigma u_\theta\right)(X_t, t, d)$ 
 $\quad t \leftarrow t + d$ 
 **end for** 
 **Return** $X_T$ |

quality, allowing for multi-step refinement when needed, thus recovering standard diffusion-based sampling. This iterative procedure is detailed in Algorithm 4.

**Approximating the normalization constant.** Beyond sample generation, a common goal in probabilistic inference is to estimate the normalization constant $Z$ of the target density. Because SCDS is formulated within the optimal control framework of stochastic sampling [7], it inherits a natural estimator of $Z$ through the Radon–Nikodym derivative between forward and backward path measures. By discretizing the likelihood ratio in (6) along simulated trajectories, SCDS enables efficient estimation of $\log Z$. In contrast, consistency-based generative models [16, 46] are trained using fully supervised losses on labeled data. As a result, they lack a built-in mechanism to estimate $Z$ or compute likelihood ratios.

# 6 Experiments

We empirically evaluate the sampling efficiency and effectiveness of the proposed CDDS and SCDS across diverse standard benchmarks in Bayesian inference and sampling [9].

Specifically, we consider Bayesian posterior inference tasks, such as Ionosphere (35-D) and the high-dimensional Log-Gaussian Cox Process (LGCP) (1600-D), alongside representative synthetic targets: a Gaussian Mixture Model (GMM) of nine components in 2-D, a 2-D Many-Well with 32 well-separated modes (MW54), the widely-studied Funnel distribution in ten dimensions, and a 50-D Many-Well distribution with 32 modes (MW52). Details are provided in Appendix

We report the Sinkhorn distance $\mathcal{W}_\gamma^2$, the Effective Sample Size (ESS), and the absolute estimation error of the log normalization constant $\Delta \log Z$ for tasks with accessible ground-truth samples. For the real-world Ionosphere and LGCP tasks, we report the evidence lower bound (ELBO). ELBO, ESS and $\Delta \log Z$ rely on importance weights. With $N > 1$ we use the discretized RND of (6); for $N = 1$ the running-cost term vanishes and the weight reduces to the boundary likelihood, which can potentially inflate ELBO scores. Hence we regard $\mathcal{W}_\gamma^2$ and $\Delta \log Z$ as primary quality indicators, and report ELBO only when ground truth samples are not available.

We benchmark our methods against established diffusion-based samplers: the Path Integral Sampler (PIS) [55], the Denoising Diffusion Sampler (DDS) [50], both trained with the KL divergence, and the Time-Reversed Diffusion Sampler (DIS) [7] trained with log-variance divergence.

Throughout our experiments, we use a pre-trained DIS as the teacher model for CDDS. Furthermore, since the optimization problem (4) may admit infinitely many solutions, we fix the noising process (3) in SCDS to ensure uniqueness of the learned solution, following the same approach as in DIS. Detailed implementation and hyperparameters are provided in Appendix A.1. The code is available at this repository.

## 6.1 Results

**Single-step sampling.** Rows shaded in in Tables 1–2 compare one–step CDDS and SCDS with fully-discretized (128–256 step) baselines. Several consistent patterns emerge. On the low–dimensional GMM and MW54 targets, CDDS attains Sinkhorn values with a factor less then double the DIS baseline. ESS is also two or three orders of magnitude larger than DIS with a single step, illustrating

that distillation corrects most of the degeneracy of naive single-step DIS. The ELBO of CDDS on Ionosphere and LGCP even exceeds that of every 256-step baseline; we attribute this to the discretized estimator of the Radon–Nikodym derivative (RND) (6), which for single-step sampling collapses to the boundary likelihood and can therefore *over-estimate* evidence.

Without access to a teacher, SCDS learns its own control and offers a trade-off: In one step SCDS delivers usable samples (e.g. GMM, Funnel) but is generally less accurate than CDDS because it has to discover the score field from scratch.

Table 1: Synthetic-benchmark results. Each task block reports the Sinkhorn distance ($\mathcal{W}_\gamma^2 \downarrow$), the effective sample size (ESS $\uparrow$), and absolute log-normalization error ($|\Delta \log Z| \downarrow$).

|  | PIS | DDS | DIS | DIS | CDDS (ours) | SCDS (ours) |
|---|---|---|---|---|---|---|
| NFE $\downarrow$ | 128 | 128 | 128 | 1 | 1 | 1 |
| **GMM (2D)** | | | | | | |
| $\mathcal{W}_\gamma^2$ | 1.7946 | 0.0898 | 0.0203 | 0.0559 | 0.0313 | 0.0478 |
| ESS | $1\times10^{-5}$ | 0.0065 | 0.8054 | 0.00057 | 0.3395 | 0.0504 |
| $|\Delta \log Z|$ | 2.1806 | 1.6819 | 0.0899 | 14.7669 | 1.5717 | 1.0743 |
| **MW54 (5D)** | | | | | | |
| $\mathcal{W}_\gamma^2$ | 0.1377 | 0.1366 | 0.1230 | 6.2804 | 0.2815 | 0.3913 |
| ESS | 0.0664 | 0.0051 | 0.2677 | $1.7\times10^{-5}$ | 0.0442 | $2.1\times10^{-5}$ |
| $|\Delta \log Z|$ | 1.9974 | 2.4154 | 1.2056 | 6766.2891 | 1.0966 | 11.2524 |
| **Funnel (10D)** | | | | | | |
| $\mathcal{W}_\gamma^2$ | 6.0731 | 5.8600 | 5.1755 | 10.5224 | 8.8849 | 5.4922 |
| ESS | $1\times10^{-5}$ | 0.0582 | 0.1305 | $4.9\times10^{-5}$ | $5.1\times10^{-5}$ | 0.00014 |
| $|\Delta \log Z|$ | 0.4381 | 0.5641 | 0.6407 | $\infty$ | 1.3316 | 10.3557 |
| **MW52 (50D)** | | | | | | |
| $\mathcal{W}_\gamma^2$ | 6.8035 | 6.7830 | 6.8808 | 31.2532 | 6.1764 | 7.1110 |
| ESS | $1\times10^{-5}$ | 0.4412 | 0.0028 | $1\times10^{-5}$ | $5.7\times10^{-5}$ | $1\times10^{-5}$ |
| $|\Delta \log Z|$ | 42.4502 | 42.4245 | 39.7814 | 9116.0713 | 63.7244 | 87.7095 |

Table 2: ELBO ($\uparrow$) on two real-world benchmarks. Shaded columns denote single–step inference.

|  | PIS | DDS | DIS | SCDS | DIS | CDDS (ours) | SCDS (ours) |
|---|---|---|---|---|---|---|---|
| NFE $\downarrow$ | 256 | 256 | 256 | 256 | 1 | 1 | 1 |
| Ionosphere (35D) | $-39.3$ | $-1510.3$ | $-77.4$ | $-73.7$ | $-3252.7$ | $-27.5$ | $-567.3$ |
| LGCP (1600D) | 397.5 | 314.8 | 365.6 | 103.9 | $-3.09\times10^6$ | 1118.0 | $-4579.7$ |

**Cost-benefit analysis of single-step inference.** Figure 2 reports the total number (training plus inference) of network-function evaluations (NFEs) for SCDS versus PIS, DDS, and DIS, that use the same N-step discretization during training. SCDS adds only three NFEs per training iteration but replaces the entire $N$-step Euler integration with a single forward pass at test time. With batch size $B$ and $I$ training iterations, the extra training cost is $3IB$ NFEs, while every sample produced at inference saves $N-1$ NFEs; hence the break-even point is $S_{\text{break}} = 3IB/(N-1)$. For the 2-D GMM experiment ($N=128$, $B=512$, $I=10{,}000$) this yields $S_{\text{break}} \approx 1.2\times10^4$. We generated five million test samples to obtain Table 1, thereby saving about 620 million NFEs relative to the baselines. In the 1600-D LGCP task the model is trained longer but with smaller batches ($N=256$, $B=64$, $I=50{,}000$), giving $S_{\text{break}} \approx 3.7\times10^4$, and generating one million samples saves over half a billion NFEs. Training cost is thus a fixed, modest overhead that is quickly amortised in realistic simulations.

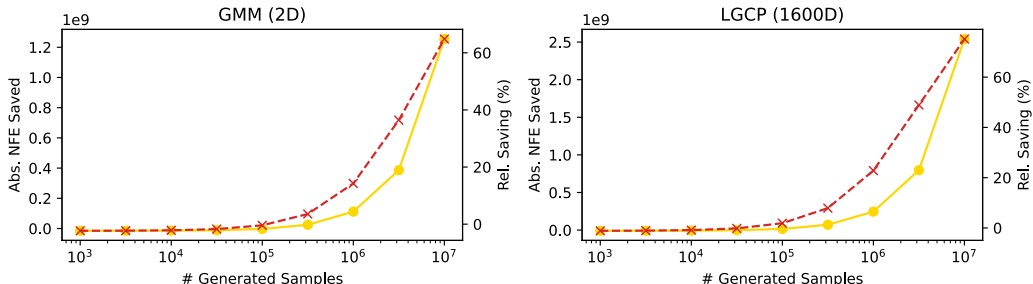

Figure 2: Absolute (yellow) and relative (red, dashed) network-function evaluations (NFE) savings of SCDS over the baseline diffusion samplers PIS, DDS, and DIS as a function of the number of generated samples (log scale). Left: 2-D GMM training regime; Right: 1600-D LGCP regime.

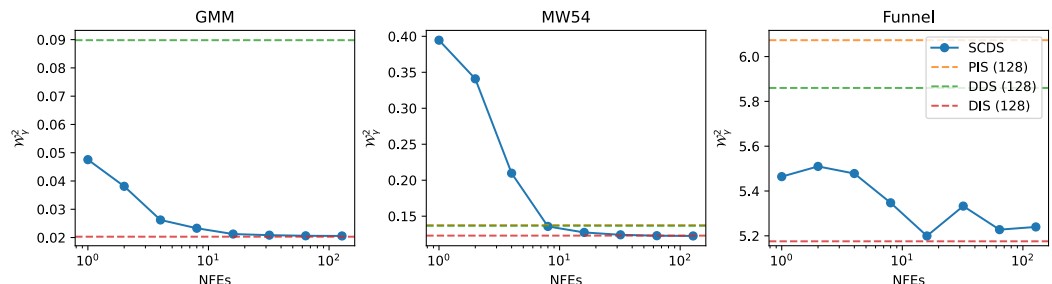

Figure 3: Sinkhorn distance as a function of the number of diffusion steps for SCDS, illustrating the flexible trade-off between computational cost and sample quality. Horizontal dashed lines represent baseline diffusion samplers with 128 steps.

Practically, we recommend using CDDS when a reliable pretrained sampler is available and the goal is single-step generation, as CDDS training is cheaper than SCDS yet could achieve comparable single-step quality. If no pretrained sampler exists, SCDS provides a more economical and flexible solution, as it learns directly from the unnormalized target and allows adjusting the number of inference steps to trade computational resources for improved sample quality.

**Trade compute for sample quality.**   Figure 3 shows how the Sinkhorn distance decays as we allocate more network-function evaluations (NFEs) to SCDS at inference time. On the 2-D GMM and the 5-D Many-Well tasks the curve is strictly monotone: with only 4–8 Euler updates SCDS already matches the accuracy of DDS, and at 32–64 steps it recovers the DIS reference obtained with 128 steps. The same trend is visible on the 10-D Funnel, but with a mild "bump" at 32 steps. SCDS allows practitioners to flexibly adjust the computational budget, progressively improving sample quality until it matches the accuracy of traditional multi-step diffusion samplers.

Additional results, comparisons with other baselines, and ablations are provided in Appendix A.3.

# 7   Conclusion

We introduced two novel approaches for efficient sampling from unnormalized target distributions: *consistency-distilled diffusion samplers* (CDDS) and the *self-consistent diffusion sampler* (SCDS). CDDS uses consistency distillation without generating a large dataset of samples. SCDS requires no pre-trained samplers and simultaneously learns to sample high-density regions and to take large steps across the path space. Our empirical results across a range of benchmarks demonstrated that both methods achieve competitive accuracy with as few as one or two steps. These findings highlighted the potential of consistency-based methods for sampling from unnormalized densities.

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
