# OpenReview forum: "Single-Step Consistent Diffusion Samplers"
_NeurIPS.cc/2025/Conference — Submitted to NeurIPS 2025_

### Official Review · Reviewer_yF5i · 2025-06-29

**Clarity:** 2
**Significance:** 2
**Originality:** 3
**Rating:** 3
**Confidence:** 3

**Summary:**

This paper extends the consistency model to the generation domain of unnormalized target distributions, and correspondingly proposes two approaches for efficient sampling from unnormalized target distributions: Consistency-Distilled Diffusion Samplers (CDDS) and Self-Consistent Diffusion Samplers (SCDS). CDDS distills a pre-trained diffusion sampler into a single-step sampler by leveraging intermediate states during the diffusion process, thereby reducing the reliance on large datasets. SCDS further eliminates the requirement for a pre-trained sampler by jointly learning diffusion transitions and large shortcut steps through a self-consistency loss. The methods are validated through extensive experiments on both synthetic and real-world benchmarks, demonstrating that they can achieve competitive sample quality with significantly reduced sampling steps.

**Questions:**

1. Could you explain the differences between the CDDS proposed in the paper and the Consistency Distillation (CD) in the original consistency models?
2. Could you elaborate on whether other diffusion acceleration techniques, such as shortcuts or consistency trajectory models, are applicable to the generation of unnormalized distributions? If so, should these state-of-the-art diffusion acceleration methods be included in the comparison?
3. The self-consistency loss in SCDS is designed to ensure that a large step yields the same result as multiple smaller steps. How sensitive is this self-consistency loss to the choice of step sizes, and are there optimal step sizes that lead to faster convergence or better sample quality?
4. Can the proposed SCDS be applied to existing image generation tasks, and what advantages does it have over existing methods?

**Ethical Concerns:**

["NO or VERY MINOR ethics concerns only"]

**Final Justification:**

I acknowledge the authors' rebuttal. They have addressed some of my concerns, and while the paper presents incremental extensions to the existing content, I will keep my original score.

**Limitations:**

yes

**Quality:**

3

**Strengths And Weaknesses:**

### Strengths:
1. The paper presents a new class of samplers that can generate high-fidelity samples in a single step, which is a significant advancement in the field of sampling from unnormalized distributions.
2. The paper provides theoretical results which shows that consistency functions can be distilled from diffusion samplers without requiring access to data from the target distribution.
3. The authors conduct extensive experiments on a variety of synthetic and real-world distributions, demonstrating the effectiveness of their methods in terms of sample quality and computational efficiency.

### Weaknesses:
1. The proposed CDDS appears to be a rather straightforward adaptation of consistency models to the generation of unnormalized distributions, which may seem somewhat incremental in terms of innovation.
2. The experimental section could benefit from comparisons to a wider range of baseline methods to further establish the superiority of the proposed approaches.

---

> ### Author Rebuttal · Authors · 2025-07-31
>
> ## Question 1
> > Could you explain the differences between CDDS and CD.
>
> We appreciate the opportunity to clarify this point further. Once a sampler is available, our consistency distillation (CDDS) approach performs consistency training similar to the original method. However, the difference is how we generate intermediate noisy states for training:
> - Generative Modeling Scenario: Intermediate state $X_{t_n}$ is generated directly by perturbing samples drawn from a given data distribution. Then, the adjacent state $X_{t_{n+1}}$ is produced by performing one Euler integration step from $X_{t_n}$ using the given pretrained sampler.
> - CDDS (our scenario): Both states are generated from simulating the pre-trained sampler from $t=0$ to $n+1$. We do not first generate a dataset by simulating the sampler up to $t=T$, and then apply the generative model scenario.
>
> Moreover, our Self-Consistent Diffusion Sampler (SCDS) further distinguishes itself from traditional consistency distillation by entirely removing the need for a pretrained sampler. Traditional consistency models rely on deterministic trajectories from data samples, providing explicit guidance towards high-density regions. In contrast, SCDS must also explore the space, which we achieve via a Brownian motion. Self-consistency is then enforced on deterministic sub-trajectories. This contrasts with the purely deterministic trajectories used in standard consistency models.
>
> We have also clarified the distinctions in the form of a table in our response to [Reviewer emRE](https://openreview.net/forum?id=eg4AmZVVPO&noteId=LiRvH8HRFJ), Question 2.
>
> ## Question 2
> > Could you elaborate on whether other diffusion acceleration techniques, such as shortcuts or consistency trajectory models, are applicable to the generation of unnormalized distributions? If so, should these state-of-the-art diffusion acceleration methods be included in the comparison?
>
> These techniques assume access to a dataset sampled from the target distribution. They lack an intrinsic mechanism to incorporate information from an unnormalized density. Our methods CDDS and SCDS precisely adapts the ideas of consistency distillation and shortcuts to the sampling problem, where only an unnormalized density is known. This difference in accessible information (also detailed in answers to Q1-3 of reviewer emRE) justifies the choice of baselines in our experiments. Please refer to [1] for an evaluation framework of diffusion samplers.
>
> ## Question 3
> > The self-consistency loss in SCDS is designed to ensure that a large step yields the same result as multiple smaller steps. How sensitive is this self-consistency loss to the choice of step sizes, and are there optimal step sizes that lead to faster convergence or better sample quality?
>
> We specifically chose step sizes that are powers of two because this minimizes the computational overhead of the self-consistency loss. The power-of-two schedule requires only 3 additional network function evaluations per training iteration. Choosing other arbitrary step sizes would increase computational cost, as intermediate points would need additional simulation.
>
> However, beyond computational convenience, the core rationale behind the self-consistency loss is not overly sensitive to the exact choice of step size as long as:
>
> - Larger steps are exactly composed of multiple smaller base steps, allowing consistent alignment of trajectories at different scales.
> - Resolution: The smallest step size ($T/N$) is sufficiently small to ensure accurate approximation of the underlying probability flow ODE.
>
> Conceptually, any choice of step sizes satisfying these criteria would ensure effective learning. Thus, while the power-of-two step size is computationally optimal and practically convenient, our method is robust and flexible enough to accommodate other step sizes without substantial differences in convergence speed or sample quality. A systematic exploration of other step-size schedules could be an interesting ablation for future work.
>
> ## Question 4
> > Can the proposed SCDS be applied to existing image generation tasks, and what advantages does it have over existing methods?
>
> SCDS is specifically designed for scenarios where the target distribution is provided through an unnormalized density. Instead, traditional image generation tasks, such as CIFAR-10 or MNIST, provide samples directly from the target distribution. We are not aware of large-scale image tasks formulated explicitly as unnormalized sampling problems [1].
>
> [1] D. Blessing, X. Jia, J. Esslinger, F. Vargas, and G. Neumann. Beyond ELBOs: A Large-Scale. Evaluation of Variational Methods for Sampling. In International Conference on Machine Learning, 2024.

---

> ### Comment · Area_Chair_DtiS · 2025-08-04
> **Action Required: Author–Reviewer Discussion Closing Soon**
>
> Dear Reviewer,
>
>
>
> This is a gentle reminder that the **Author–Reviewer Discussion** phase ends within just three days (by **August 6**). Please take a moment to read the authors’ rebuttal thoroughly and engage in the discussion. Ideally, every reviewer will respond so the authors know their rebuttal has been seen and considered.
>
>
>
> Thank you for your prompt participation!
>
>
>
> Best regards,
>
>
>
> AC

---

> ### Comment · Area_Chair_DtiS · 2025-08-05
> **Action Required: Author–Reviewer Discussion Closing Soon**
>
> Dear Reviewer,
>
> A gentle reminder that the extended Author–Reviewer Discussion phase ends on **August 8 (AoE)**.
>
> Please read the authors’ rebuttal and participate in the discussion **ASAP**. Regardless of whether your concerns have been addressed, kindly communicate:
>
> - If your concerns have been resolved, please acknowledge this clearly.
>
> - If not, please communicate what remains unaddressed.
>
> The “Mandatory Acknowledgement” should only be submitted **after**:
>
> - Reading the rebuttal,
>
> - Engaging in author discussion,
>
> - Completing the "Final Justification" (and updating your rating).
>
> **As per policy, I may flag any missing or unresponsive reviews and deactivate them once additional reviewer feedback has been posted.**
>
> Thank you for your timely and thoughtful contributions.
>
>
>
>
> Best regards,
>
> AC

---

> ### Comment · Area_Chair_DtiS · 2025-08-06
> **Action Required: Author–Reviewer Discussion Closing Soon**
>
> Dear Reviewer,
>
> Please notice that the “Mandatory Acknowledgement” should only be submitted **after**:
>
> - Reading the rebuttal,
>
> - Engaging in author discussion,
>
> - Completing the "Final Justification" (and updating your rating).
>
> Kindly review the authors’ response and reply as soon as possible. Regardless of whether your concerns have been fully addressed:
>
> - If your concerns are resolved, kindly confirm this clearly.
>
> - If not, please explain what issues remain.
>
> Thank you for your cooperation.
>
>
> Best,
>
> AC

---

> ### Comment · Reviewer_yF5i · 2025-08-07
>
> I acknowledge the authors' rebuttal. They have addressed some of my concerns, and while the paper presents incremental extensions to the existing content, I will keep my original score.

---

> > ### Author Response · Authors · 2025-08-08
> >
> > Thank you for the follow‑up. We appreciate that some of your concerns were addressed. To help us improve the final version, could you please indicate which specific concerns remain unaddressed?
> >
> > **On the experimental coverage you flagged as weak.**
> > We are extending the evaluation to include DIS / DDS / PIS / SCDS on a 40‑mode Gaussian mixture with long diffusion chains (as in Blessing et al., 2024, Table 10). These results will be added to the camera‑ready. We will also report wall‑clock timing on an NVIDIA RTX A6000 for both training and inference, alongside the existing NFE counts already reported in the paper.
> >
> > **On the “incremental” nature of CDDS/SCDS relative to consistency models.**
> > We agree that our CDDS draws inspiration from consistency training; however, our contribution is to apply and prove its viability in the setting of unnormalized target densities, i.e., without access to data from the targe, by distilling from a sampler’s probability‑flow ODE states rather than from a dataset. This is different from standard consistency generative models and, to our knowledge, has not been shown before in this sampling context.
> >
> > For SCDS, the novelty is twofold:
> > - Step‑size–conditioned control. We condition the control $u_\theta(X_t, t, d)$ on the desired step size $d$, unifying small diffusion steps and large jumps within one network. This lets us train end‑to‑end and fully amortize exploration during training, enabling single‑step inference with optional few‑step refinement at test time.
> > - Scaled score component for unnormalized densities. The architecture explicitly includes a learned scaling of the analytic score $\nabla \log \rho(X_t)$, which departs from the standard consistency function used in generative modeling and is tailored to sampling from unnormalized targets (Appendix A.1).
> >
> > Together, these design choices address the core challenge of no data from the target: we learn to explore with the SDE and to take large, self‑consistent shortcuts with the PF‑ODE without a dataset or teacher, while keeping training overhead modest (SCDS adds only three extra function evaluations per iteration; Sec. 5.2 / Alg.2).
> >
> >
> > Planned additions to the camera‑ready:
> > - Results on the 40‑mode GMM with long chains (DIS/DDS/PIS/SCDS), including metrics consistent with our current tables.
> > - Wall‑clock runtime on RTX A6000 for training and inference, reported next to NFE counts (Appendix A.3 already documents the hardware we used).
> > - A brief clarification table contrasting our CDDS/SCDS with standard consistency models in generative settings (data‑driven) vs. our unnormalized‑density setting (density‑oracle‑driven).
> >
> > If there are specific baselines or ablations you would like to see, please let us know we’re happy to incorporate them.
> >
> > Thank you again for the careful review and for helping us make the paper stronger.

---

### Official Review · Reviewer_MBBz · 2025-07-01

**Clarity:** 3
**Significance:** 2
**Originality:** 2
**Rating:** 3
**Confidence:** 4

**Summary:**

This paper introduces two diffusion sampling methods for unnormalized target distributions. Consistency-Distilled Diffusion Samplers (CDDS) leverage a pretrained diffusion sampler to perform single-step sampling via consistency distillation. Self-Consistent Diffusion Samplers (SCDS) learn to sample in a single step from scratch, combining score estimation and consistency learning within a unified training objective, and requiring no pretrained model. The paper evaluates their efficiency and effectiveness across a range of synthetic and real-world sampling benchmarks.

**Questions:**

- Could the authors clarify which specific discrepancy measure is used in Equation (7)?
 - How do the proposed methods compare to the baselines in terms of computational capacity, such as the number of parameters? This is important for fair comparisons.
 - How strong/realistic is the assumption of a Lipschitz constraint in Theorem 1, especially in the SDE setting where involved quantities may not be differentiable? Additionally, how does the method account for or bound the error when the corresponding loss function exceeds zero?

**Ethical Concerns:**

["NO or VERY MINOR ethics concerns only"]

**Final Justification:**

I thank the authors for their responses. However, the issues I raised regarding the novelty of CDDS and the experimental performance of SCDS remain insufficiently addressed.

**Limitations:**

Yes

**Quality:**

2

**Strengths And Weaknesses:**

## Strengths:
 - The paper addresses the important problem of enhancing the sampling efficiency of diffusion samplers. Improvements in this area have the potential to substantially benefit a wide range of real-world applications that rely on efficient sampling methods.
 - The manuscript is well-organized and clearly written. The exposition is concise yet thorough, facilitating a clear understanding of the core contributions and methodologies.

## Weaknesses:
 - CDDS requires both a pretrained sampler and additional training specific to CDDS, which introduces computational overhead. This cost must be amortized by generating a large number of inference samples to be justifiable.
 - The conceptual novelty of CDDS appears limited. It builds upon the same consistency principle as used in Consistency Models (Song, 2023), and Theorem 1 is borrowed from that work. While the application to sampling is different, the methodological innovation is relatively incremental.
 - The empirical results, particularly for SCDS on real-world benchmarks, are not consistently compelling. To more convincingly demonstrate the advantages of the proposed method, a broader experimental evaluation is needed. This should include:
   - Real-world benchmarks such as Bayesian inference tasks.
   - Synthetic tasks with increased multimodality and higher dimensionality (where longer diffusion chains may better explore the space).
   - Detailed analysis of how reduced NFEs translate into actual computational (time) savings (both at training and inference time).
   - Investigation into how much the performance can be improved by increasing the number of sampling steps until the computational budget is matched with compared baselines.
   - Provide error bars for experimental results.

---

> ### Author Rebuttal · Authors · 2025-07-31
>
> ## Question 1
> > Could the authors clarify which specific discrepancy measure is used in Equation (7)?
>
> The discrepancy measure used in Equation (7) is a standard mean squared error. We have corrected (7) accordingly.
>
> ## Question 2
> > How do the proposed methods compare to the baselines in terms of computational capacity, such as the number of parameters? This is important for fair comparisons
>
> In our experiments, all baselines including PIS, DIS, DDS, and our proposed CDDS are of all the same size. The network is composed of a MLP with four layers and 64 units per layer along with a Fourier-based time embedding of the same dimensions. SCDS has an additional small 2 layers time-embedding to condition on the step-size. Therefore, all methods have a similar computational capacity and the comparisons are fair. We’ve included these details and discussions about the number of parameters in the revised manuscript.
>
> ## Question 3
> > How strong/realistic is the assumption of a Lipschitz constraint in Theorem 1, especially in the SDE setting where involved quantities may not be differentiable? Additionally, how does the method account for or bound the error when the corresponding loss function exceeds zero?
>
> We thank the reviewer for this insightful question. The Lipschitz continuity assumption in Theorem 1 applies to the learned consistency function defined as a deterministic mapping from states along the PF ODE. The Lipschitz continuity assumption in Theorem 1 is a standard condition used widely in the analysis of neural ODE (e.g., [1]). Typically, neural networks with standard activations and bounded parameters satisfy this constraint.
>
> Theorem 1 proves the network can be arbitrarily accurate if the loss reaches zero, which has provided sufficient support of the theoretical correctness of our method. When the corresponding loss function exceeds zero gradually, we intuitively believe the sample quality will drop. Nonetheless, it could still perform well with some non-zero losses, as empirically verified by our experiments. For instance, single-step CDDS beats PIS and DDS with 128 steps in terms of Sinkhorn distance for the GMM experiment. CDDS also recovers all modes in one step as shown in Figure 4. Theoretically, deriving a precise and quantitative error bound as a function of the non-zero loss value is challenging, as it may be influenced by a complex interplay of factors like the non-convex nature of the loss landscape, the specific local minimum found by the optimization algorithm, and the generalization properties of the network, which cannot be captured by the single loss value alone. As the primary contribution of our paper is the proposal of single-step samplers, we believe this detailed analysis on error bound of non-zero loss scenarios is beyond the scope of this paper.
>
> [1] Y. Song, P. Dhariwal, M. Chen, and I. Sutskever. Consistency Models. In International Conference on Machine Learning, 2023.

---

> > ### Comment · Reviewer_MBBz · 2025-08-07
> >
> > I appreciate the authors’ response. However, the core issues I identified regarding the novelty of the work and the adequacy of the experimental results remain insufficiently resolved. Therefore, I will maintain my original score.

---

> > > ### Author Response · Authors · 2025-08-08
> > >
> > > Thank you for your continued engagement and for outlining the precise points that still concern you. Below we clarify the empirical strength, novel contributions, and computational trade‑offs of our work, and we specify the additional experiments and plots that will appear in the camera‑ready version.
> > >
> > > ## Real‑world Bayesian benchmarks
> > > We already evaluate SCDS on two challenging posterior‑inference tasks.
> > >
> > > * **Ionosphere (35D):** Bayesian logistic‑regression.
> > > * **Log‑Gaussian Cox Process (LGCP, 1600D):** a popular high-dimensional task in spatial statistics which models the position of pine saplings.
> > >
> > > On single‑step inference SCDS greatly outperforms (in terms of ELBO) the one‑step DIS baseline and, on Ionosphere, even beats DDS‑256:
> > > | task | DIS‑1 | SCDS‑1 | improvement factor|
> > > |------|-------------|-------------------|-------------|
> > > | Ionosphere | $−3252.7$ | $−567$ | $5.7$ |
> > > | LGCP | $−3.09 \times 10^6$ | $−4.6 10^3$ | $670$ |
> > >
> > > ## Novelty beyond plain consistency distillation
> > >
> > > The critical novelty and innovation of our methods, especially compared to standard consistency distillation, arise explicitly from the fundamental difference in the available information:
> > >
> > > - Standard consistency distillation methods assume direct access to samples drawn from the target distribution, allowing straightforward perturbations for training.
> > > - CDDS, in contrast, does not rely on target samples. Instead, it adapts consistency distillation specifically to the unnormalized-density setting, utilizing partial integrations of the probability flow ODE from the pretrained sampler itself, a setting unexplored in prior work.
> > >
> > > Moreover, our second method, SCDS, is even more innovative, as it entirely removes the dependency on a pretrained sampler by jointly learning both the sampler and consistency conditions from scratch, explicitly leveraging the unnormalized density directly throughout the entire training process.
> > >
> > > For further clarification, we summarized the distinctions explicitly in a detailed table provided in our response to reviewer emRE ([available here](https://openreview.net/forum?id=eg4AmZVVPO&noteId=iLFVbNf65m)). We will include a concise but clear comparison table in the final version of the manuscript.
> > >
> > > ## Broader evaluation and statistical rigour
> > >
> > > We are currently training DIS/DDS/PIS/SCDS on a 40‑mode Gaussian mixture with long diffusion chains (Blessing et al., Tab.10). The results will be included. We will also include the actual wall-clock timing (RTX A6000) for training + inference.
> > >
> > > Std bars already are included in Appendix Table 4-5 and will be moved to the main body.
> > >
> > > Moreover, Figure 3 already shows that SCDS’ Sinkhorn distance generally monotonically decreases as we allocate more steps, matching the baselines step‑for‑step. In contrast, the performance of multi-step sampling of traditionnal consistency models generally reaches a plateau quickly (<5 steps) without matching the baselines.
> > >
> > > We hope this clarifies your concern. Please let us know if further details would help resolve any remaining concerns.
> > >
> > > D. Blessing, X. Jia, J. Esslinger, F. Vargas, and G. Neumann. Beyond ELBOs: A Large-Scale. Evaluation of Variational Methods for Sampling. In International Conference on Machine Learning, 2024

---

> > > > ### Comment · Reviewer_MBBz · 2025-08-09
> > > >
> > > > I thank the authors for their additional response. However, my concerns regarding the novelty of CDDS and the experimental performance of SCDS could not be resolved.

---

> ### Comment · Area_Chair_DtiS · 2025-08-04
> **Action Required: Author–Reviewer Discussion Closing Soon**
>
> Dear Reviewer,
>
>
>
> This is a gentle reminder that the **Author–Reviewer Discussion** phase ends within just three days (by **August 6**). Please take a moment to read the authors’ rebuttal thoroughly and engage in the discussion. Ideally, every reviewer will respond so the authors know their rebuttal has been seen and considered.
>
>
>
> Thank you for your prompt participation!
>
>
>
> Best regards,
>
>
>
> AC

---

> ### Comment · Area_Chair_DtiS · 2025-08-05
> **Action Required: Author–Reviewer Discussion Closing Soon**
>
> Dear Reviewer,
>
> A gentle reminder that the extended Author–Reviewer Discussion phase ends on **August 8 (AoE)**.
>
> Please read the authors’ rebuttal and participate in the discussion **ASAP**. Regardless of whether your concerns have been addressed, kindly communicate:
>
> - If your concerns have been resolved, please acknowledge this clearly.
>
> - If not, please communicate what remains unaddressed.
>
> The “Mandatory Acknowledgement” should only be submitted **after**:
>
> - Reading the rebuttal,
>
> - Engaging in author discussion,
>
> - Completing the "Final Justification" (and updating your rating).
>
> **As per policy, I may flag any missing or unresponsive reviews and deactivate them once additional reviewer feedback has been posted.**
>
> Thank you for your timely and thoughtful contributions.
>
>
>
>
> Best regards,
>
> AC

---

> > ### Comment · Area_Chair_DtiS · 2025-08-06
> > **Action Required: Author–Reviewer Discussion Closing Soon**
> >
> > Dear Reviewer,
> >
> > This is, **again**, a reminder to actively participate in the author–reviewer discussion (please also refer to the previous two reminders, **Action Required: Author–Reviewer Discussion Closing Soon**).
> >
> > **Per policy, failure to respond to the authors’ rebuttal may result in your review being flagged as insufficient, which could lead to desk rejection of your own submission and be recorded for future reference.**
> >
> > Please read the authors’ response and reply as soon as possible. Regardless of whether your concerns have been fully addressed:
> >
> > - If your concerns are resolved, kindly confirm this clearly.
> >
> > - If not, please explain what issues remain.
> >
> > Thank you for your cooperation.
> >
> > Best,
> >
> > AC

---

### Official Review · Reviewer_Z1sa · 2025-07-02

**Clarity:** 2
**Significance:** 2
**Originality:** 3
**Rating:** 3
**Confidence:** 4

**Summary:**

This paper introduces a consistency model approach for sampling from unnormalized distributions using only one (or a few) steps. The authors propose two variants: (1) CCDS, which distills an existing trained diffusion-based sampler into a consistency model; and (2) SCDS, which trains a consistency model from scratch using a combination of a sampling loss and a self-consistency loss.

Compared to existing diffusion-based samplers, the proposed methods reduce the number of steps required to generate target samples. Additionally, as the proposed approach is grounded in the optimal control framework, it is able to estimate the normalization constant of the target distribution.

Both CCDS and SCDS are benchmarked against established diffusion-based samplers on a range of synthetic and real-world Bayesian posterior inference tasks. While CCDS appears to have reasonable compute-accuracy trade-offs, the results for SCDS are a bit mixed.

**Questions:**

- Eq 2,3: what are $\pi$ and $\tau$? Are they supposed to be the diffusion prior $p_\text{prior}$ and target $p_\text{target}$ mentioned in the text above?

- Eq 2, 3: which one is the reverse  (generative) process and which one is the forward process? From the following definition of $\mu_\theta$ as a learned control, it appears that Eq2 corresponds to the reverse process. Could the authors confirm?

- Eq 6: $u$ should be $u_\theta$?  And $p_\text{prior}, p_\text{target}$ should be $\pi$ and $\tau$?

- The definition of consistency function in line 117 of page 3 seems to be fairly arbitrary. Is the consistency function $f$ in Theorem 1 uniquely defined? And what does it mean for "consistency function of a probability flow ODE"?

- Instead of using the large datasets, CCDS integrates partial ode. Can this trick be applied to standard consistency model training?

- Eq 7, should be a minus sign in between functions?

- The comment after Theorem 1 says "consistency functions can be distilled from diffusion samplers when only an unnormalized density oracle is available" -- I wonder where the unnormalized density is used for distillation?

- In practice, how is the sampling loss in eq 5 computed?

- Does estimating the log normalization constant, or ESS,  require discretizing the diffusion chain over multiple timesteps? In that case, is NFE of CCDS or SCDS still 1?

- The discretized RND should be explained in more details.

- In appendix line 813 of page 21, "v=0" is assumed -- why this is the case?

- On page 7 line 248-249: is N the number of sampling steps? What is the "running-cost term"?

- Table 2 results: the authors may also consider computing the posterior predictive log likelihood on a held-out test dataset.

- Could the authors provide a clearer summary of the empirical performance of CCDS and SCDS, especially considering statistical significance?

**Ethical Concerns:**

["NO or VERY MINOR ethics concerns only"]

**Final Justification:**

After discussing with the authors, I have decided to maintain my current rating. While the paper presents some interesting ideas, the novelty seems to be limited. The writing clarify can be improved, and the experimental validation could be made more thoroug, particularly by including comparisons to other single-step baseline methods and evaluating with more meaningful metrics.

**Limitations:**

The paper does not appear to include a discussion of its limitations. While SCDS does not seem to yield consistently strong empirical results, CCDS also has an important limitation: it requires a pretrained diffusion sampler, which may itself be inaccurate or costly to train. A potential avenue for improvement could be to incorporate a sampling loss into CCDS, which may help correct deficiencies in the pretrained model during distillation.

**Quality:**

2

**Strengths And Weaknesses:**

Strengths:

- The paper is well-motivated with the goal to address the challenge of reducing the number of steps required in diffusion-based samplers.

- The proposed consistency model framework is both intuitive and appealing, offering a promising direction for efficient sampling.

- Empirically, CCDS demonstrates reasonable compute-accuracy trade-offs and even occasionally outperforms diffusion baselines.

Weakness:

- The writing could be significantly improved. In particular, the background on consistency models should be made more technical, especially given how frequently the concept is referenced. This would also help clarify the novelty of applying consistency models in the sampling setting. Additionally, the probability flow ODE, which central to the proposed method, is referenced multiple times but never written out explicitly. There are also typos and unclear sentences throughout (see "Questions" section for examples).

- The empirical advantages of CCDS and SCDS remain unclear. Results lack standard errors, and statistically significant differences are not clearly highlighted. SCDS appears to underperform in several cases, especially on the ELBO metric in Table 2.

- Limited novelty for the CCDS approach as it seems to  a straightforward application of consistency distillation to a pre-trained diffusion sampler.

- Several related works on single-step samplers are missing from the discussion and empirical comparison, e.g. [1,2]/

[1] Efficient and unbiased sampling of boltzmann distributions via consistency models
Fengzhe Zhang, Jiajun He, Laurence I Midgley, Javier Antorán, José Miguel Hernández-Lobato

[2] Training Neural Samplers with Reverse Diffusive KL Divergence
Jiajun He, Wenlin Chen, Mingtian Zhang, David Barber, José Miguel Hernández-Lobato

---

> ### Author Rebuttal · Authors · 2025-07-31
>
> **Question 1:**
> > In Eq. 2,3, what are $\pi$ and $\tau$? Are they supposed to be the diffusion prior $p_{\text{prior}}$ and target $p_{\text{target}}$?
>
> Yes, $\pi$ and $\tau$ represent the initial probability distributions for the forward-time process and reverse-time process, respectively. As clarified explicitly in the manuscript (lines 93-94), by selecting $\pi = p_{\text{prior}}$ and $\tau = p_{\text{target}}$ in our setting, we generate samples from the target distribution $p_{\text{target}}$ using Eq. (2) with the optimal control $u^*$. We have clarified this in the revised version.
>
> **Question 2:**
> > Which of Eq. 2 or 3 is the reverse (generative) process, and which is the forward process?
>
> In our notation, Eq. (2) indeed corresponds to the generative (forward-time) process.  Note that in the diffusion sampling literature [1], the generative process (from prior to target) is conventionally called the forward process because the noising process cannot be directly simulated from data. The two processes are time-reversals of each other.
>
> **Question 3:**
> > In Eq. 6, should $P^{u, p_{\text{prior}}}$ and $P^{v, p_{\text{target}}}$ appear explicitly instead of $P^{u,\pi}$ and $\overline{P}^{v,\tau}$?
>
> Thank you for pointing this out. We have updated Eq. (6) to remain general, explicitly using the general symbols $u, v, \pi, \tau$ for clarity and consistency, and we removed the parameter superscript to avoid confusion:
>
> $$ \log\frac{\mathrm d {P}^{u,\pi} }{\mathrm d \overline{P}^{v,\tau}} = \int_0^T (u + v) \cdot \Big(u + \frac{v - u}{2} + \nabla \cdot (\sigma v - \mu)\Big)\mathrm d s + \int_0^T (u + v) \mathrm d W_s +  \log \frac{\pi (X_0)}{\tau(X_T)}$$
>
> **Question 4:**
> > Is the consistency function $f(X_t, t)$ uniquely defined in Theorem 1, and what does it mean for it to be a ‘consistency function of a probability flow ODE’?
>
> Yes, for any given initial condition (sample from the prior), the Probability Flow ODE is deterministic, thus uniquely defining the consistency function as the deterministic mapping from $X_t$ to $X_T$. The consistency function of the PF ODE, therefore, refers to the unique deterministic mapping induced by integrating the PF ODE forward in time.
>
> **Question 5:**
> > Can the partial ODE integration used in CCDS be applied to standard consistency model training?
>
> In principle, yes, although we have not empirically evaluated it. However, we anticipate greater integration errors compared to standard consistency training, which typically benefits from exact perturbation kernels that provide effectively ground-truth intermediate noisy data (assuming the drift is linear in X).
>
> **Question 6:**
> > Should there be a minus sign between functions in Eq. 7?
>
> Yes, thank you for pointing out this typo. We have corrected Eq. (7) accordingly
>
> **Question 7:**
> > Where is the unnormalized density oracle used in the distillation process (CDDS)?
>
> The unnormalized density is indirectly utilized during training of the teacher sampler, which guides the intermediate trajectories used for distillation. As you correctly suggest, an interesting future improvement could be to incorporate a sampling loss directly into the distillation step (CDDS) to potentially correct deficiencies of the pretrained sampler during distillation.
>
> **Question 8:**
> > How is the sampling loss (Eq. 5) computed in practice?
>
> The sampling loss (KL or LV divergence) is computed explicitly using discretization of the Radon–Nikodym derivative. Algorithm 5 in the appendix provides detailed steps for this computation. We will explicitly mention that the detailed steps are provided in Algorithm 5 in the revision.
>
> **Question 9:**
> > Does estimating log normalization constant or ESS require multiple diffusion steps, thus increasing NFE beyond 1?
>
> When performing single-step sampling (CDDS/SCDS), the NFE is still one. Specifically, the interior integral term collapses leaving a single evaluation of the likelihood ratio at the boundary points only.
>
> **Question 10:**
> > The discretized Radon–Nikodym derivative should be explained in more detail.
>
> In practice, we approximate the integrals in the Radon–Nikodym derivative by replacing them with finite sums over a discrete time mesh. The implementation of the discretized Radon–Nikodym derivative is provided in Appendix Section A.1, under the paragraph “Discretizing the Radon–Nikodym derivative.”
>
> **Question 11:**
> > In Appendix line 813, why is $v=0$ assumed?
>
> This assumption is made explicitly to ensure uniqueness of the solution to Eq. (4). Without fixing the noising process (the backward process defined by $v$), multiple solutions may exist [1]. We have explicitly stated our choice of a Variance-Preserving SDE as the noising process in the revised manuscript.
>
> **Question 12:**
> > In lines 248–249, is $N$ the number of sampling steps, and what is the running-cost term?
>
> Yes, $N$ is the number of sampling steps; we have clarified this explicitly in the revised manuscript. The running-cost term corresponds to the integral over time in Eq. (6), and, when discretized, corresponds to the sum computed in line 12 of Algorithm 5. We have clarified this terminology when first introducing the RND (Eq. 6).
>
> **Question 13:**
> > Have the authors considered computing posterior predictive log likelihood on a held-out test dataset for Table 2?
>
> For Table 2, we compute only the ELBO since we do not have direct access to real samples from these targets. Posterior predictive log likelihood would require samples or predictions on held-out data, which is not available in these tasks as formulated.
>
> **Question 14:**
> > Could the authors provide a clearer summary of the empirical performance of CCDS and SCDS, especially considering statistical significance?
>
> CDDS consistently provides strong single-step sampling performance. SCDS, while slightly underperforming in some high-dimensional scenarios, still achieves strong performance, demonstrating its capability to learn effective single-step sampling without pretrained samplers. We have included standard deviations explicitly in Appendix Tables 4 and 5 to strengthen the empirical claims further.
>
>
> [1] Lorenz Richter and Julius Berner. Improved sampling via learned diffusions. In International Conference on Learning Representations, 2024.

---

> > ### Comment · Reviewer_Z1sa · 2025-08-05
> >
> > I would like to thank the authors for their detailed rebuttal. Most of my questions are answered. However, there are a few unresolved concerns and new followup questions.
> >
> > 1. Several related works on single-step samplers are missing from the discussion and empirical comparison, e.g. [1,2].
> >
> > 2. Follow-up on question 7 and concern on limited novelty:
> >
> > Is my understanding correct that the current method only distills from an existing teacher diffusion sampler, and the target density information is only utilized in training the teacher, not in the distillation stage? If that is the case, is the proposed approach essentially consistency distillation applied to a diffusion model (which happens to target at unnormalized distribution). And how would you summarize the novelty of this approach?

---

> > > ### Comment · Reviewer_Z1sa · 2025-08-05
> > >
> > > I also wanted to add a followup to question 13: while the held-out data is not available for problems in Table 2, I would encourage the authors to evaluate the proposed method in bayesian posterior inference task where held-out test data are often available, for example, see posterior db benchmark https://github.com/stan-dev/posteriordb.

---

> > > > ### Author Response · Authors · 2025-08-07
> > > >
> > > > **Concern on Related Works.**
> > > > We agree that a broader coverage of diffusion-acceleration techniques developed for generative modeling will help situate our contributions more clearly. In the camera-ready version we will add an extended Related-Work section to the Appendix that explicitly mentions these methods.
> > > >
> > > > **Follow-up on Question 7 and concern on limited novelty.**
> > > > Your understanding is correct regarding CDDS: our method distills consistency from a pretrained diffusion sampler, and the target density is leveraged indirectly through the pretrained sampler during distillation. However, the innovation of our methods arise from the fundamental difference in the available information:
> > > > - Standard consistency distillation methods assume direct access to samples: the intermediate state $X_{t_n}$ is generated directly by perturbing samples drawn from a given data distribution and a tractable Gaussian perturbation kernel. Then, the adjacent state $X_{t_{n+1}}$ is produced by performing one Euler integration step from $X_{t_n}$ using the given pretrained model.
> > > > - In our scenario, both states are generated from simulating the pre-trained sampler from $t=0$ to $t=t_{n+1}$. We do not first generate a dataset by simulating the sampler up to $t_N$, and then apply the generative model scenario.
> > > >
> > > > Moreover, our Self-Consistent Diffusion Sampler (SCDS) further distinguishes itself from traditional consistency distillation by entirely removing the need for a pretrained sampler. Traditional consistency models rely on deterministic trajectories from data samples, providing explicit guidance towards high-density regions. In contrast, SCDS must also explore the space, which we achieve via a Brownian motion. Self-consistency is then enforced on deterministic sub-trajectories. This contrasts with the purely deterministic trajectories used in standard consistency models.
> > > >
> > > > For further clarification, we summarized the distinctions explicitly in a detailed table provided in our response to reviewer emRE ([available here](https://openreview.net/forum?id=eg4AmZVVPO&noteId=iLFVbNf65m)).
> > > >
> > > > **Follow-up on Question 13.**
> > > > Your proposal to benchmark on PosteriorDB is well-taken. We concentrated on the same diffusion-sampling tasks that prior samplers use (for instance, DIS [1], DDS [2], PIS [3], and Blessing et al. [4]) so that the improvements in NFE and quality are directly comparable to the established literature.
> > > >
> > > > Running a rigorous PosteriorDB study requires substantial per-model hyper-parameter tuning, which is beyond what we can complete before the rebuttal deadline.  We will explicitly list PosteriorDB evaluation as future work and intend to release results in a follow-up appendix / code update in an archive version.
> > > >
> > > > [1] Berner, Richter, Ullrich (2024). An Optimal Control Perspective on Diffusion-based Generative Modeling. Transactions on Machine Learning Research.
> > > >
> > > > [2] F. Vargas, W. S. Grathwohl, and A. Doucet. Denoising Diffusion Samplers. In International Conference on Learning Representations, 2023.
> > > >
> > > > [3] Q. Zhang and Y. Chen. Path Integral Sampler: A Stochastic Control Approach For Sampling. In International Conference on Learning Representations, 2022.
> > > >
> > > > [4] D. Blessing, X. Jia, J. Esslinger, F. Vargas, and G. Neumann. Beyond ELBOs: A Large-Scale Evaluation of Variational Methods for Sampling. In International Conference on Machine Learning, 2024.

---

> ### Comment · Area_Chair_DtiS · 2025-08-04
> **Action Required: Author–Reviewer Discussion Closing Soon**
>
> Dear Reviewer,
>
>
>
> This is a gentle reminder that the **Author–Reviewer Discussion** phase ends within just three days (by **August 6**). Please take a moment to read the authors’ rebuttal thoroughly and engage in the discussion. Ideally, every reviewer will respond so the authors know their rebuttal has been seen and considered.
>
>
>
> Thank you for your prompt participation!
>
>
>
> Best regards,
>
>
>
> AC

---

### Official Review · Reviewer_NinP · 2025-07-03

**Clarity:** 2
**Significance:** 3
**Originality:** 3
**Rating:** 4
**Confidence:** 1

**Summary:**

This paper introduces **single-step diffusion samplers** that generate samples from unnormalized distributions without requiring iterative SDE simulation. Two methods are proposed:

1. **Consistency-Distilled Diffusion Samplers (CDDS)**: distills a pre-trained sampler into a single-step function using consistency training, even without full trajectories.
2. **Self-Consistent Diffusion Samplers (SCDS)**: learns a single-step sampler **from scratch**, combining traditional sampling loss with a novel **self-consistency loss** across time scales.

These methods amortize inference-time costs into training, yielding significant speedups while maintaining high-quality sampling. Extensive experiments are conducted on both synthetic and real-world unnormalized distributions.

**Questions:**

- Why should the self-consistency objective work? Why does this trick on 'synthetic' samples from the control function work? If there would be theoretical/intuitive/toy problem justification for this, the paper would be a lot stronger, as SCDS is the main contribution of the paper.
- Why would the SCDS method not degrade the sample quality? Perhaps there are some theoretical results you could draw on regarding samplers not perfectly recovering the reverse trajectory (prior to target) but still having good performance.
-  Can CDDS or SCDS be used for image generation tasks? If not, clarify limitations. Showing results on CIFAR-10 or MNIST would broaden impact.

**Ethical Concerns:**

["NO or VERY MINOR ethics concerns only"]

**Final Justification:**

The review has been adequately addressed. Furthermore, given the other discussions by other reviewers of the significance of the experiments, this reviewer recommends a borderline accept.

**Limitations:**

No, negative societal impacts were not addressed. How could these samplers be negatively affect society?

**Quality:**

2

**Strengths And Weaknesses:**

Strengths:
- Dramatic reduction in inference cost is highly relevant
- Extends consistency training to **unnormalized sampling** tasks
- encouraging theoretical results for CDDS training
- encouraging experimental results on small benchmarks

Weaknesses:
- Most results focus on 2D synthetic or small-scale benchmarks; scalability to image generation is not demonstrated
- Lack of theoretical grounding and/or motivation on the key innovation for SCDS samplers

---

> ### Author Rebuttal · Authors · 2025-07-31
>
> ## Question 1
> > Why should the self-consistency objective work? Why does this trick on ‘synthetic’ samples from the control function work? If there would be a theoretical/intuitive/toy problem justification for this, the paper would be a lot stronger, as SCDS is the main contribution of the paper.
>
> The self-consistency objective intuitively works because it explicitly enforces that taking a single large step from an intermediate noisy state should yield the same outcome as multiple smaller steps. Specifically, as detailed in the second paragraph of Section 5.2, the model is trained such that predictions of larger steps (step size 2d) match the results from sequences of smaller incremental steps (two steps of size d).
>
> Initially, the network learns a reliable “base” transition for the smallest step size $d = T/N$, guided by the stochastic diffusion process from the unnormalized density $\rho$. This base transition helps discover high-density regions (Section 5.2, first paragraph). For larger step sizes, the model’s predicted state is matched against states obtained by recursively applying previously learned smaller-step transitions. Hence, the model uses its own “synthetic” states (with gradients stopped) as targets to enforce consistency at progressively larger scales.
>
> To better highlight this intuition, we have refined the Overview Section 5.1:
>
> > Even though CDDS demonstrates that single-step sampling is feasible, it requires a pretrained diffusion sampler for distillation. In this section, we further introduce *self-consistent diffusion sampler* (SCDS) that achieves single-step sampling without requiring a pre-trained diffusion sampler. The intuition behind SCDS is to reconcile the conflicting objectives of diffusion-based sampling and consistency functions. Diffusion-based samplers typically learn a time-dependent control function designed to steer an SDE from the prior toward the target through multiple small steps. In contrast, consistency models aim to directly map an intermediate noisy state from an ODE trajectory to the final target state.
>
> > We propose a unified network architecture that conditions the control function $u_\theta(X_t, t, d)$ simultaneously on the current time $t$ and the desired step size $d$. By varying the step size parameter $d$, the model smoothly transitions between small, incremental steps (as in diffusion-based samplers) and large leaps (as in consistency models). For smaller steps $d = T/N$, the model learns stochastic incremental diffusion-based sampling guided by the density $\rho$, allowing exploration of the target distribution. For larger step sizes, the model is trained such that its outcomes match those obtained from sequences of smaller steps, thereby enforcing consistency across different scales. This step-conditioned control architecture amortizes the learning of both small transitions and large jumps within a single network. During inference, setting $d = T - t$ enables single-step sampling directly from any intermediate noisy state, whereas setting $d = T/N$ recovers traditional multi-step diffusion sampling.
>
> ## Question 2:
> > Why would the SCDS method not degrade the sample quality? Perhaps there are some theoretical results you could draw on regarding samplers not perfectly recovering the reverse trajectory (prior to target) but still having good performance.
>
> Previous work [1] has formally derived the Radon–Nikodym derivative (RND) of Equation (6). If the RND is driven to 1, the forward and backward path measures become identical. In practice, perfect recovery of these reverse trajectories is not necessary to obtain high-quality samples. Like ours, multiple previous works such as [2] have obtained good results minimizing discretized versions of (6).
>
> Intuitively, the self-consistency loss doesn't conflict with Radon–Nikodym derivative because our network models a step-size-conditioned control (scaled score) instead of a consistency function. By conditioning the on the step size, our architecture accommodates both short-step diffusion updates (required for accurate RND estimation) and larger shortcut steps across the Euler integration. Consequently, this unified modeling approach does not compromise but rather supports accurate estimation of the RND, and facilitates recursive learning of single-step sampling.
>
> ## Question 3:
> > Can CDDS or SCDS be used for image generation tasks? If not, clarify limitations. Showing results on CIFAR-10 or MNIST would broaden impact.
>
> We appreciate this insightful suggestion. However, CDDS and SCDS as presented are specifically designed for sampling from unnormalized target densities, where no direct samples are available. Commonly used image datasets such as CIFAR-10 or MNIST inherently provide samples rather than unnormalized densities. We are not aware of standardized unnormalized-density large image generation tasks. We refer to [2] for an evaluation framework and standard sampling benchmark.
>
> ## Question 4:
> > Negative societal impacts were not addressed. How could these samplers negatively affect society?”
>
> While we do not anticipate major negative societal impacts from our sampling methods themselves, though we acknowledge potential misuse of efficient sampling algorithms in sensitive applications, such as synthetic data generation or privacy. We’ve updated the Limitations section in the Appendix.
>
> [1] J. Berner, L. Richter, and K. Ullrich. An Optimal Control Perspective on Diffusion-based Generative Modeling. Transactions on Machine Learning Research, 2024.
>
> [2] D. Blessing, X. Jia, J. Esslinger, F. Vargas, and G. Neumann. Beyond ELBOs: A Large-Scale Evaluation of Variational Methods for Sampling. In International Conference on Machine Learning, 2024.

---

> ### Comment · Area_Chair_DtiS · 2025-08-04
> **Action Required: Author–Reviewer Discussion Closing Soon**
>
> Dear Reviewer,
>
>
>
> This is a gentle reminder that the **Author–Reviewer Discussion** phase ends within just three days (by **August 6**). Please take a moment to read the authors’ rebuttal thoroughly and engage in the discussion. Ideally, every reviewer will respond so the authors know their rebuttal has been seen and considered.
>
>
>
> Thank you for your prompt participation!
>
>
>
> Best regards,
>
>
>
> AC

---

> > ### Comment · Area_Chair_DtiS · 2025-08-05
> > **Action Required: Author–Reviewer Discussion Closing Soon**
> >
> > Dear Reviewer,
> >
> > A gentle reminder that the extended Author–Reviewer Discussion phase ends on **August 8 (AoE)**.
> >
> > Please read the authors’ rebuttal and participate in the discussion **ASAP**. Regardless of whether your concerns have been addressed, kindly communicate:
> >
> > - If your concerns have been resolved, please acknowledge this clearly.
> >
> > - If not, please communicate what remains unaddressed.
> >
> > The “Mandatory Acknowledgement” should only be submitted **after**:
> >
> > - Reading the rebuttal,
> >
> > - Engaging in author discussion,
> >
> > - Completing the "Final Justification" (and updating your rating).
> >
> > **As per policy, I may flag any missing or unresponsive reviews and deactivate them once additional reviewer feedback has been posted.**
> >
> > Thank you for your timely and thoughtful contributions.
> >
> >
> >
> >
> > Best regards,
> >
> > AC

---

> > > ### Comment · Area_Chair_DtiS · 2025-08-06
> > > **Action Required: Author–Reviewer Discussion Closing Soon**
> > >
> > > Dear Reviewer,
> > >
> > > This is, **again**, a reminder to actively participate in the author–reviewer discussion (please also refer to the previous two reminders, **Action Required: Author–Reviewer Discussion Closing Soon**).
> > >
> > > **Per policy, failure to respond to the authors’ rebuttal may result in your review being flagged as insufficient, which could lead to desk rejection of your own submission and be recorded for future reference.**
> > >
> > > Please read the authors’ response and reply as soon as possible. Regardless of whether your concerns have been fully addressed:
> > >
> > > - If your concerns are resolved, kindly confirm this clearly.
> > >
> > > - If not, please explain what issues remain.
> > >
> > > Thank you for your cooperation.
> > >
> > > Best,
> > >
> > > AC

---

> ### Comment · Reviewer_NinP · 2025-08-06
>
> The reviewer thanks the author for their response. The intuition behind the self-consistency objective is appreciated.
>
> Furthermore, while the reviewer does not have the background to adjudicate on the significance of the experiments, the responses to other reviewers has been helpful.
>
> This reviewer will bump the score up by one. However, for a stronger score, some kind of theoretical result, perhaps on a simplified problem, is necessary with respect to SCDS.

---

### Official Review · Reviewer_emRE · 2025-07-10

**Clarity:** 4
**Significance:** 2
**Originality:** 3
**Rating:** 5
**Confidence:** 3

**Summary:**

The paper propose to extend on-step consistency diffusion samplers that "convert"/distill computationally intensive diffusion samplers (computationally expensive at inference because they require a lot of ODE/SDE steps) into computationally efficient one-step samplers.
These consistency models already existed, and the paper extends this idea to obtain get efficient unnormalized distribution samplers.

**Questions:**

- Can authors explain in which way the proposed method is new? and not "just" distill a sampler, as in the original consistency sampler?
- Maybe it would help to clarify the following sentence: "Notably, unlike standard consistency generative models, the states $\hat{X}_{t_n + 1}$ and $\hat{X}_{t_n }$ are obtained from partial integrations of the probability flow ODE rather than from real data samples.". Could you provide a clear step-by-step comparison of consistency models for generative models and for unnormalized distribution sampling?
- Can you also comment on the following? " While our distillation approach builds upon the core principles of consistency generative models, it differs in setting and requirements. **Instead of relying on having access to a dataset from ptarget , our method extends consistency distillation to sampling from unnormalized distributions, making it applicable beyond generative modeling tasks.**" The way I understand this last sentence is that the proposed method only requires a sampler (whether trained on data or unnormalized density), but once you have the sampler, the distillation is the same.
If this is not the same, could you please provide a detailed explanation of why this is not the case?

I am not familiar with the standard experiments in the sampling literature, and I am not able to asses how challenging/significative the experimental results are. It would help if authors could comment on it.

I am very willing to change my score if authors provide a clear answer to my main concern

**Ethical Concerns:**

["NO or VERY MINOR ethics concerns only"]

**Final Justification:**

The reviewers answered all my concerned, hence I raised my score.

However, I do not consider myself an expert in the field of 'sampling' from an energy function: I have trouble assessing how challenging are the experiments, and how new is the contribution.
Hence, I lowered my confidence.

**Quality:**

3

**Strengths And Weaknesses:**

Strength:
- The paper is very clear, Figure 1 is the clearest explanation of diffusion models I have ever seen.
- Experiments show one-step sampling with only mild performance decrease

Weaknesses:
- I do not understand the novelty of the paper: once one has a sampler one can distill it with a consistency loss, whether the sampler is built to sample from unnormalized data distribution, or sample from an empirical data distribution

---

> ### Author Rebuttal · Authors · 2025-07-31
>
> ## Question 1
> > Can authors explain in which way the proposed method is new? and not “just” distill a sampler, as in the original consistency sampler?
>
> Our approach indeed builds upon the principles of consistency models introduced by [1], but differs in terms of the information available. Original consistency models were developed for generative modeling, assuming access to a dataset sampled from a target distribution. This enables easy generation of intermediate states via known perturbation kernels from these real samples.
>
> In our problem formulation, we have no direct access to samples from the target distribution, only an unnormalized density oracle. Consequently, we cannot generate noisy intermediate states from target samples. To overcome this, we introduce two distinct methods tailored to the sampling setting:
>
> - Consistency-Distilled Diffusion Sampler (CDDS): This method distills a pretrained sampler for unnormalized distributions. Critically, we leverage noisy states obtained directly from intermediate partial integrations of the sampler from the prior. It entirely removes the need for data from the target or datasets of fully diffused samples.
> - Self-Consistent Diffusion Sampler (SCDS): Our second approach is even more novel. Since there is no data to reveal high-density regions, SCDS simultaneously explores high-density regions via Brownian motion and learns to skip intermediate Euler steps. To our knowledge, SCDS is the first attempt to combine consistency with stochastic control-based diffusion samplers to achieve single-step sampling without external pretrained models or data.
>
> Our methods extend the scope of consistency models to sampling scenarios that were previously unexplored.
>
> ## Question 2
> > Could you provide a clear step-by-step comparison of consistency models for generative models and for unnormalized distribution sampling?
>
> Yes, we provide a comparison table which clearly compares the differences between existing consistency distillation and consistency training methods, with our proposed CDDS and SCDS:
> | Aspect                             | Consistency Distillation | Consistency Training | CDDS (ours) | SCDS (ours) |
> |------------------------------------|----------------------|----------------------|-------------|-------------|
> | **Access to data samples**         | Yes                  | Yes                  | No          | No          |
> | **Access to unnormalized density** | No                   | No                   | Yes         | Yes         |
> | **Pretrained sampler required?**   | Yes | No | Yes | No |
> | **Form of learned model**          | Consistency function $f_\theta(X_t, t) \mapsto X_T$ | Consistency function $f_\theta(X_t, t) \mapsto X_T$ | Consistency function $f_\theta(X_t, t) \mapsto X_T$ | Step-size conditioned scaled score model $u_\theta(X_t, t, d)$|
> | **Training objective**             | Consistency regression between adjacent noisy states | Consistency regression between adjacent noisy states | Consistency regression between adjacent noisy states | Joint optimization of sampling loss from forward SDE and self-consistency loss on deterministic transitions |
> | **Noisy states generation** | From real data samples via $p(X_t\|X_T)$ with $X_T \sim p_{\text{target}}$ | From real data samples via $p(X_t\|X_T)$ with $X_T \sim p_\text{target}$ | From partial integration of the pretrained sampler's from $X_0 \sim p_\text{prior}$ | From integration of the forward process using $u_\theta(X_t, t, d)$ |
>
> We’ve added this concise comparison table to the revised manuscript.
>
> ## Question 3
> > Regarding the sentence "While our distillation approach builds upon the core principles…" it seems the proposed method only requires a sampler, and once you have the sampler, the distillation is the same. Is this correct? If not, please clarify.
>
> We appreciate the opportunity to clarify this point further. Once a sampler is available, our consistency distillation (CDDS) approach performs consistency training similar to the original method. However, the difference is how we generate intermediate noisy states for training:
> - Generative Modeling Scenario: Intermediate state $X_{t_n}$ is generated directly by perturbing samples drawn from a given data distribution. Then, the adjacent state $X_{t_{n+1}}$ is produced by performing one Euler integration step from $X_{t_n}$ using the given pretrained sampler.
> - CDDS (our scenario): Both states are generated from simulating the pre-trained sampler from $t=0$ to $n+1$. We do not first generate a dataset by simulating the sampler up to $t=T$, and then apply the generative model scenario.
>
> Moreover, our Self-Consistent Diffusion Sampler (SCDS) further distinguishes itself from traditional consistency distillation by entirely removing the need for a pretrained sampler. Traditional consistency models rely on deterministic trajectories from data samples, providing explicit guidance towards high-density regions. In contrast, SCDS must also explore the space, which we achieve via a Brownian motion. Self-consistency is then enforced on deterministic sub-trajectories. This contrasts with the purely deterministic trajectories used in standard consistency models.
>
> ## Question 4
> > Can you comment on the significance/challenge of the experimental benchmarks used?
>
> We use a range of well-established and challenging benchmarks standard in the Bayesian inference and sampling literature, such as the Gaussian mixture model, funnel distribution, many-well, log-Gaussian Cox process (1600 dimensions), and Bayesian logistic regression on the ionosphere dataset representing realistic inference problems of considerable complexity and scale.
> Our results show that our methods significantly accelerate sampling via learned diffusion (1 compared to 128 sampling steps). We refer to [2] for an evaluation framework of diffusion-based samplers.
>
>
> [1] Y. Song, P. Dhariwal, M. Chen, and I. Sutskever. Consistency Models. In International Conference on Machine Learning, 2023.
>
> [2] D. Blessing, X. Jia, J. Esslinger, F. Vargas, and G. Neumann. Beyond ELBOs: A Large-Scale Evaluation of Variational Methods for Sampling. In International Conference on Machine Learning, 2024.

---

> > ### Comment · Reviewer_emRE · 2025-08-04
> >
> > I thank the author for the in-depth rebuttal.
> > I have one remaining question before making my final decision:
> > In Table 2, why the 'distilled' model CDDS has much higher ELBO than the non-distilled counterparts?
> > Should'nt the 'distilled' be slightly worst?

---

> > > ### Author Response · Authors · 2025-08-06
> > >
> > > This is an important question. The key reason behind the observed ELBO discrepancy relates to how we discretize the Radon–Nikodym derivative (RND) when using divergence-based (line-integral) formulations.
> > >
> > > **Background on the RND and ELBO**
> > > We discretize the forward–backward RND as a line integral (Eq. 6 in our paper). The Riemann sum is described in Algorithm 5. This approach follows directly from Girsanov’s theorem, which expresses the RND as a stochastic integral plus a deterministic integral term involving the controls.
> > >
> > > **Why the line-integral discretization doesn't guarantee a strict ELBO bound?**
> > > When discretized as a numerical approximation (e.g., Euler–Maruyama), this integral representation does not inherently guarantee a strict lower bound on the log-normalization constant. The reason is that the discretization introduces numerical errors especially prominent for single-step approximations. This phenomenon was also observed in prior work on diffusion samplers and optimal control formulations, such as [1] and [2], and discussed in [3].
> > >
> > > **Our recommendations and primary metrics**
> > > Because of these known discretization effects, we explicitly mention in the paper (lines 250–251) that we regard Sinkhorn distances and absolute `log Z` estimation error as our primary quality indicators. The ELBO metric should be viewed with caution and interpreted alongside other metrics.
> > >
> > > We agree this point warrants explicit clarification, and we have now included these detailed explanations and references explicitly in the revised manuscript.
> > >
> > > [1] Berner, Richter, Ullrich (2024). An Optimal Control Perspective on Diffusion-based Generative Modeling. Transactions on Machine Learning Research.
> > >
> > > [2] Richter & Berner (2024). Improved Sampling via Learned Diffusions. ICLR.
> > >
> > > [3] Blessings, Berner, Richter, Neumann (2025). Underdamped Diffusion Bridges with Applications to Sampling. ICLR.

---

> > > > ### Comment · Reviewer_emRE · 2025-08-07
> > > >
> > > > I thank the authors for the in-depth rebuttal.
> > > > They answered all my questions and I raised my score

---

> ### Comment · Area_Chair_DtiS · 2025-08-04
> **Action Required: Author–Reviewer Discussion Closing Soon**
>
> Dear Reviewer,
>
>
>
> This is a gentle reminder that the **Author–Reviewer Discussion** phase ends within just three days (by **August 6**). Please take a moment to read the authors’ rebuttal thoroughly and engage in the discussion. Ideally, every reviewer will respond so the authors know their rebuttal has been seen and considered.
>
>
>
> Thank you for your prompt participation!
>
>
>
> Best regards,
>
>
>
> AC

---

### Comment · Area_Chair_DtiS · 2025-08-01
**Kindly Engage with Author Responses**

Dear Reviewers,





The authors have submitted their responses to your reviews. At your earliest convenience, please take a moment to engage with their replies. Your continued discussion and clarifications will be invaluable in ensuring a fair and constructive review process for all parties.

Thank you again for your thoughtful contributions and dedication.





Warm regards,



Your AC

---

### Decision · Program_Chairs · 2025-09-17

**Decision:**

Reject

**Comment:**

This paper extends ideas from consistency and shortcut models to the challenging task of sampling from unnormalized distributions without access to target samples. However, multiple reviewers (e.g., yF5i, Z1sa, MBBz) raised concerns that the contribution is incremental with limited novelty. I therefore recommend **rejection**.